# Assessment of Compaction, Temperature, and Duration Factors for Packaging and Transporting of Sterile Male *Aedes aegypti* (Diptera: Culicidae) under Laboratory Conditions

**DOI:** 10.3390/insects13090847

**Published:** 2022-09-17

**Authors:** Beni Ernawan, Tjandra Anggraeni, Sri Yusmalinar, Hadian Iman Sasmita, Nur Fitrianto, Intan Ahmad

**Affiliations:** 1Institut Teknologi Bandung (ITB), School of Life Sciences and Technology, Jalan Ganesha No. 10, Bandung 40132, Indonesia; 2Research Center for Radiation Process Technology, Research Organization for Nuclear Energy, National Research and Innovation Agency of Indonesia (BRIN), Jalan Lebak Bulus Raya No. 49, Jakarta 12440, Indonesia

**Keywords:** *Aedes aegypti*, packaging, transportation, chilling temperature, compaction

## Abstract

**Simple Summary:**

The sterile insect technique (SIT) is a proven method to control some insect pests and is currently being tested to control some mosquito species, including *Aedes aegypti*. It is challenging to maintain the quality of sterile male mosquitoes in operational SIT during the packaging and transportation processes. The experiment presented in this manuscript was undertaken to investigate compaction, temperature, and duration factors during the packaging and transportation of gamma-sterilized male *Ae. aegypti*. The effects of packaging and transportation factors on the quality parameters of gamma-sterilized male *Ae. aegypti*—mortality, flight ability, induced sterility, and longevity—were assessed. The results of this experiment demonstrate appropriate packaging and transportation conditions for maintaining the quality of gamma-sterilized male *Ae. aegypti*.

**Abstract:**

Optimized conditions for the packaging and transportation of sterile males are crucial factors in successful SIT programs against mosquito vector-borne diseases. The factors influencing the quality of sterile males in packages during transportation need to be assessed to develop standard protocols. This study was aimed to investigate the impact of compaction, temperature, and duration factors during packaging and transportation on the quality of gamma-sterilized male *Ae. aegypti*. *Aedes aegypti* males were sterilized at a dose of 70 Gy, compacted into Falcon tubes with densities of 40, 80, and 120 males/2 mL; and then exposed to temperatures of 7, 14, 21, and 28 °C. Each temperature setup was held for a duration of 3, 6, 12, 24, and 48 h at a 60 rpm constant vibration to simulate transportation. The parameters of mortality, flight ability, induced sterility, and longevity were investigated. Results showed that increases in density, temperature, and duration significantly increased mortality and reduced flight ability and longevity, but none of the factors significantly affected induced sterility. With a mortality rate of less than 20%, an escaping rate of more than 70%, considerable longevity, and the most negligible effect on induced sterility (approximately 98%), a temperature of 7 °C and a compaction density of 80 males/2 mL were shown to be optimized conditions for short-term transportation (no more than 24 h) with the minimum adverse effects compared with other condition setups.

## 1. Introduction

*Aedes aegypti* (Linnaeus) is considered one of the most dangerous animals on the planet due to its vectorial capacity for major human diseases, including dengue, chikungunya, yellow fever, and Zika [1,2]. Historically, this species originated from Africa (*Aedes aegypti formosus*) and then distributed worldwide (*Aedes aegypti aegypti*), especially in tropical areas [3,4,5]. Nevertheless, the authors of a recent study found that the ancestor of *Ae. aegypti* in the African continent originates from the Aegypti Group from islands in the southwestern Indian Ocean [6]. The spread of *Ae. aegypti*, as well as the vectorial diseases, are rapidly causing a global health burden [7]. Since effective vaccines and preventive drugs are lacking, vector population control plays an essential role against the diseases transmitted by *Ae. aegypti* [8]. Conventional control methods, such as reducing breeding site density and applying insecticide have been used to reduce dengue cases. However, no satisfactory trend has resulted [9]. Approximately 3.83 billion people (around 53% of the global population) live in suitable dengue risk areas, including Asia, Central America, and Central Africa [7,10]. Thus, efficient, sustainable, and environmentally friendly control methods are urgently needed [11]. One such method is the radiation-based sterile insect technique (SIT). This method is environmentally friendly, target-specific, and can be combined with other vector control methods in area-wide integrated pest management (AW-IPM) [12,13].

For several decades, the SIT has been successfully applied to eradicate some major insect pests, including the New World screwworm *Cochliomyia hominivorax* (Coquerel) and the Mediterranean fruit fly *Ceratitis capitata* (Wiedemann) in America, the melon fly *Bactrocera cucurbitae* (Coquillett) in Japan, and the tsetse fly *Glossina austeni* in Tanzania [14,15,16]. In the past, SIT trials were implemented to control several mosquito species, including *Aedes* sp., *Anopheles* sp., and *Culex* sp. However, no satisfying result was reported [17,18]. In the last ten years, improvements, including in equipment and procedures, have been made in SIT programs for mosquitoes toward operational levels [19]. Several pilot SIT implementations have been reported to reduce *Ae. aegypti* and *Ae. albopictus* populations in various parts of the world, with promised outcomes of between 70 and 90% population suppression [20,21,22,23,24,25,26].

An inevitable limitation of SIT is that it causes a reduction in sterilized male competitiveness, potentially due to colonization, sex separation, sterilization, packaging, transporting, and release methods [27]. The authors of several studies reported overcoming this limitation through the development and standardization of colonization methods, including protocols regarding equipment, artificial larval diet, and the rapid quality control of flight ability [28,29,30,31,32,33,34]. Previous studies have shown the critical factors that influence mosquito sterilization when using gamma irradiation, including irradiation dose, pupal age, and oxygen level, which have been used to develop a standard sterilization protocol [35,36,37,38,39,40]. Despite several studies reporting optimized conditions for packing and transporting sterile male mosquitoes, no standard protocol is available for handling, packaging, transporting, and releasing sterile male mosquitoes, especially *Ae. aegypti*.

Previous studies have developed the informative baseline in mosquito packaging and transporting, showing that temperature and compaction factors significantly impact the survival of several male mosquito species, including *Ae. aegypti*, *Ae*. *albopictus,* and *An*. *arabiensis* [41,42,43]. For example, Chung et al. [41] reported the best conditions for packaging and transporting *Ae. aegypti*, i.e., at 7 °C and at a density of 40 male mosquitoes/cm^3^. Results indicated that low temperature and compaction with a specific density reduced the mortality of the sterile male mosquito. However, a comprehensive investigation of the effects of temperature, compaction, duration, and gamma irradiation treatments on the quality of sterile male *Ae. aegypti* during handling, packaging, and transporting is not yet available.

In this study, we investigated the combined effects of handling, packaging, and transporting gamma-sterilized male *Ae. aegypti*, including temperature, compaction, and duration. We assessed the following sterile male *Ae. aegypti* quality parameters: mortality, longevity, flight ability, and induced sterility. This study provides additional comprehensive information that can be used to develop a standard protocol for handling, packaging, and transporting sterile male mosquitoes in SIT programs.

## 2. Materials and Methods

### 2.1. Mosquito Strain

The *Ae. aegypti* strain used for our experiments originated from field collection in South Tangerang City, Banten Province, Indonesia, and has been maintained at the Research Center for Radiation Process Technology-National Research and Innovation Agency of Indonesia (BRIN), Jakarta, since 2017. The colony was maintained at a climate-controlled insectary at a temperature of 26 ± 2 °C, relative humidity (RH) of 70 ± 10%, and photoperiod of 12:12 h. The mosquito strain maintenance procedure was described in detail by Ernawan et al. [40].

### 2.2. Gamma Irradiation Procedure

The gamma irradiator used in the experiments was a Gammacell model 220 (originally manufactured by Atomic Energy of Canada Ltd., Ottawa, Canada, in 1968, upgraded by Institute of Isotopes, Co., Ltd., Budapest, Hungary, in 2015) with a cobalt-60 (Co-60) source (current activity of 4870 Curie and dose rate of 3514 Gy/h on 14 July 2021) located in the Research Center for Radiation Process Technology-BRIN, Jakarta. Routine dosimetry calibration was conducted and resulted in approximately 3% uncertainty of the absorbed dose (certificate no. 19C-109B, accredited by DTU Nutech, Denmark, 2019; certificate no. N° ID C/ET 23-11/1075, accredited by Aerial, France, 2021). Male *Ae. aegypti* pupae were placed into a transparent plastic tube (14 cm in diameter and 3 cm high), excess water was removed (pupae remained damp), and they were irradiated at a dose of 70 Gy [40].

### 2.3. Temperature Regime in Packed Conditions

Irradiated male *Ae. aegypti* pupae were placed into the adult cage and supplied with a 10% (*v*/*v*) sucrose solution. One day post-emergence, *Ae. aegypti* males were anesthetized at a temperature of 4 °C for approximately 5 min [44], and then 40, 80, and 120 males were counted and transferred into 15 mL Falcon tubes (Biologix Plastic Changzhou Co., Ltd., Jinan, Shandong, China) and compacted by pressing a 1.5 cm × 1.5 cm × 1.5 cm sponge down to 2 mL to achieve individual densities of 40, 80, and 120 males/2 mL, respectively. The cap and the bottom end of the tube were drilled to create holes using a 1.5 mm spiral drill needle (Model DIN 338 R-N, Guhring, Germany) to allow for airflow [41]. Tubes were transferred into a 1 L beaker glass and then placed into a water bath shaker (Model OLS26, Grant Instruments, Ltd., Cambridge, UK) set to temperatures of 7, 14, 21, and 28 °C. Each temperature setup was performed for 3, 6, 12, 24, and 48 h at a constant vibration of 60 rpm to simulate transportation. Both unirradiated–unpacked and irradiated–unpacked specimens maintained under laboratory conditions were used as the control. Three replicates were carried out for each combination of factors. The studied parameters were mortality, flight ability, induced sterility, and longevity.

### 2.4. Data Collection

#### 2.4.1. Mortality Rate

To examine the effects of packaging and simulated transportation on mortality, irradiated male *Ae. aegypti* inside the Falcon tubes in each treatment were transferred into a cage (17.5 cm × 17.5 cm × 17.5 cm, Bugdorm-4M1515, MegaView Science Co., Ltd., Taichung, Taiwan) and provided with a 10% (*v*/*v*) sucrose solution. At 24 h post-treatment, the mortality rate was determined by dividing the dead specimens by the initial numbers of each treatment.

#### 2.4.2. Flight Ability

Male *Ae. aegypti* in each treatment were tested for their flight ability according to the method of Bond et al. [37] with a slight modification. Briefly, all *Ae. aegypti* males inside each Falcon tube at each density and treatment were poured into a Petri dish (9 cm in diameter) equipped with a transparent plastic tube (8 cm in diameter and 25 cm high). This equipment was placed inside a 160 cm × 160 cm × 180 cm insect tent (Bugdorm-2960 insect rearing tent, MegaView Science Co., Ltd., Taichung, Taiwan). Flight ability was determined according to the proportion of escapes over a 24 h period.

#### 2.4.3. Induced Sterility

For each treatment, 20 treated sterile *Ae. aegypti* males were randomly selected and allowed to mate with 20 unmated females (1:1 ratio) in a cage (30 cm × 30 cm × 30 cm) with continuous access to a 10% (*v*/*v*) sucrose solution. Meanwhile, for the control, 20 unirradiated males were allowed to mate with 20 unmated females (1:1 ratio). After a 3-day mating period, females were provided a sheep’s blood meal. Three days post-blood feeding, females were allowed to oviposit in a filter-paper-lined plastic cup. Egg paper was collected and slow-dried over four days under laboratory conditions for maturation before hatching. The egg hatching rate was determined by observing the detached operculum under a stereomicroscope (Model SMZ 745, Nikon Corp., Minato-ku, Tokyo, Japan). Residual fertility was determined as the percentage of control fertility, and induced sterility was determined by subtracting 100% from residual fertility [39].

#### 2.4.4. Longevity under Laboratory Conditions

To investigate the effects of packaging and simulated transportation on longevity, 75 *Ae. aegypti* males were randomly selected from each treatment and evenly distributed into three cages (17.5 cm × 17.5 cm × 17.5 cm, Bugdorm-4M1515, MegaView Science Co., Ltd., Taichung, Taiwan). Each cage was continuously supplied with a 10% (*v*/*v*) sucrose solution. Control unirradiated–unpacked and irradiated–unpacked specimens were maintained under laboratory conditions. Longevity was determined by recording the survival (interval of 24 h) until all males succumbed to natural mortality.

### 2.5. Statistical Analysis

Prior to the statistical analysis, data were transformed using arcsine square root (sqrt) and tested for normality and homogeneity. General linear model (GLM) full univariate factorial followed by post hoc Tukey test was used to analyze the influence of the treatments on the parameters of mortality, flight ability, and induced sterility. A Kaplan–Meier survival analysis followed by Mantel–Cox log-rank test was used to analyze longevity in different treatments. Statistical analysis was performed using the Statistical Package for the Social Sciences (SPSS) version 22 for Windows (International Business Machine Corp., Armonk, NY, USA).

## 3. Results

### 3.1. Mortality Rate

The statistical analysis showed that mortality of gamma-sterilized male *Ae. aegypti* was significantly affected by density, temperature, and duration as a single factor. In the two-way interaction, there was no interaction effect on the mortality rate between duration and density, while the effects of duration and temperature differed depending on the level of temperature and density, respectively. Furthermore, in the three-way interaction, the effects density, temperature, and duration differed from the simple sum of their effects (Table 1). The range of mortality percentages, across the different levels of duration and density, varied among the four temperature levels: 5.0 ± 2.6% to 32.5 ± 6.29%, 13.33 ± 3.63% to 38.89 ± 2.82%, 7.5 ± 1.44% to 87.5 ± 6.29%, and 4.58 ± 0.83% to 86.94 ± 3.38% for the temperature levels of 7, 14, 21, and 28 °C, respectively (Table 2). Interactions between the three factors could be seen at lower temperatures (7 and 14 °C), at which mortality showed no difference among all densities at durations from 3 to 24 h then significantly increased at a duration of 48 h. However, at higher temperatures (21 and 28 °C, mortality showed no difference among all densities at durations from 3 to 12 h and from 3 to 6 h at temperatures of 21 and 28 °C, respectively, then significantly increased with increasing duration. In general, mortality significantly increased with increasing density, temperature, and duration of the packaging and simulated transportation treatment. Additionally, the effect of duration on the mortality rate did not differ across the levels of density (df = 8, F = 1.799, *p* = 0.082) (Table 1); this phenomenon was mainly seen at lower temperatures (7 and 14 °C) (Table 2); however, at higher temperatures (21 and 28 °C) and longer durations (12 to 48 h), mortality significantly increased at the highest density (120 males/2 mL).

### 3.2. Flight Ability

The flight ability of gamma-sterilized male *Ae. aegypti* showed a significant reduction compared with both unirradiated–unpacked and irradiated–unpacked controls, except for the duration of 3 h at lower temperatures (7 and 14 °C) (Table 3). The flight ability parameter was significantly affected by a three-way interaction between density, temperature, and duration (df = 24, F = 2.239, *p* = 0.001) (Table 1). Mean flight ability ranged from 23.44 ± 3.8% to 100.0 ± 0.0%, 34.06 ± 3.76% to 93.13 ± 3.59%, 23.13 ± 4.75% to 82.5 ± 5.1%, and 28.54 ± 0.02% to 80.31 ± 0.04% at temperatures of 7, 14, 21, and 28 °C, respectively (Table 3). Lower temperatures (7 and 14 °C) could maintain the flight ability of more than 70% in all durations except for 48 h. However, keeping the males at 21 °C for, at most, 12 h, and at 28 °C for 3 h resulted in escape rates of more than 70% (Table 3). Generally, flight ability was significantly reduced with increasing density, temperature, and duration.

### 3.3. Induced Sterility

The density, temperature, duration, and their interaction were negligible factors affecting induced sterility (df = 20, F = 0.474, *p* = 0.972) (Table 1). The mean induced sterility in all treatments ranged from 95.34 ± 2.27% to 99.52 ± 0.24% and indicated no significant differences (Table 4).

### 3.4. Longevity under Laboratory Conditions

The survival probability of gamma-sterilized male *Ae. aegypti* was reduced by all factors of the treatments, i.e., density, temperature, and duration (Figure 1). The mean longevity of the treatment groups was significantly reduced compared with both unirradiated–unpacked and irradiated–unpacked controls. Among the packaging and simulated transportation treatments, the mean longevity ranged from 3.05 ± 0.3 to 6.87 ± 0.43 days, 3.23 ± 0.39 to 6.72 ± 0.49 days, 1.48 ± 0.14 to 7.77 ± 0.49 days, and 1.43 ± 0.14 to 8.04 ± 0.73 days at temperatures of 7, 14, 21, and 28 °C respectively (Appendix A). In general, longevity was significantly reduced with increasing density, temperature, and duration of the treatment (log-rank test, *p* < 0.05) (Appendix A). In addition, longevity at densities of between 40 and 80 males/2 mL with lower temperatures (7 and 14 °C) and the same duration showed no significant difference, except for durations of 12 h and 48 h at temperatures of 7 and 14 °C, respectively.

## 4. Discussion

Handling, packaging, and transporting male insects are critical steps in SIT programs. Considering the fragility of mosquitoes, it is challenging to maintain the quality of the sterile male specimens during handling, packaging, and transport. The main objective of packaging and transportation development is to construct appropriate methods associated with easy handling, space efficiency, and suitable conditions that can maintain the quality of the sterile male mosquitoes in long-term transportation. The authors of the present study investigated key factors that potentially influence the packaging and transporting of gamma-sterilized male *Ae. aegypti*, i.e., compaction, temperature, and duration, at the laboratory scale. In addition, we simulated packaging and transportation; subsequently, we observed the sensitivity of sterile male *Ae. aegypti* to the packaging and transportation factors of treatments i.e., mortality, flight ability, longevity, and induced sterility.

In the present study, the mortality rate of gamma-sterilized male *Ae. aegypti* in all treatments ranged from 5.0 ± 2.6% to 87.5 ± 6.29% and showed a tendency to significantly increase with increases in density, temperature, and duration of the packaging and simulated transportation. However, no difference was found between the densities of 40 and 80 males/2 mL (Table 2), so the density of 80 males/2 mL is better from a performance point of view. We found that a temperature of 7 °C and a density of 80 males/2 mL comprised the optimal setup with the smallest effect on the mortality of the gamma-sterilized male *Ae. aegypti*, which was less than 20% at all durations (Table 2). These findings are close to those of previous studies, which showed that temperatures ranging from 7 to 10 °C during transportation resulted in maintaining the survival of *Ae. aegypti* [41,43]. Regarding species, previous studies showed that *Ae*. *albopictus* also have a similar tolerance to temperatures from 7 to 10 °C for packing and transportation [43,45]. In addition, *An*. *arabiensis* could be immobilized at a temperature ranging from 4 to 10 °C for durations of up to 24 h without any significant adverse impacts [42]. Our findings are also close to those resulting from the temperature treatments used to maintain Mediterranean fruit flies [46] and tsetse fly *Glossina palpalis gambiensis* [47] for long-distance transportation (around 10 °C).

Flight ability is one of the essential quality attributes of sterile male insects in SIT programs. This parameter is associated with the ability of sterile males to disperse, survive, find food, and seek and mate wild-type females in the field, which determines the success of SIT implementation [48]. In this study, the flight ability of gamma-sterilized male *Ae. aegypti* was significantly affected by all treatment factors, i.e., density, temperature, and duration. In general, flight ability was considerably reduced with increases in density, temperature, and duration. However, no difference was found between densities of 40 and 80 males/2 mL. We found that a temperature of 7 °C and a density of 80 males/2 mL with a duration of no more than 24 h comprised the optimal setup which could maintain the flight ability from 78.13 ± 1.49% to 100.00 ± 0.0% (Table 3). A similar result was reported by Mastronikolos et al. [45], who transported irradiated male *Ae*. *albopictus* from Italy to Greece by compacting them at temperatures of 8 to 14 °C. They found that this temperature treatment during transportation could maintain the flight ability by more than 60%. In addition, the chilled temperature treatment during packaging and transportation was expected to lead to an immobilization state, reducing interaction and friction among individual gamma-sterilized male mosquitoes inside the compaction packing device. Such friction may lead the physical damage, including missing scales, head damage, abdomen damage, wing damage, antenna damage, and leg damage that potentially reduce flight ability [41].

Regarding induced sterility, we found that the factors during packaging and simulated transportation treatments, i.e., density, temperature, and duration, did not affect male *Ae*. *aegypti’s* sterility. Induced sterility ranged from 95.34 ± 2.27% to 99.52 ± 0.24%; however, there was no statistical difference (Table 4). In the present study, based on our previous study, the gonad cells of male *Ae. aegypti* were exposed to gamma irradiation at a dose of 70 Gy (Cobalt-60), therefore causing dominant lethal mutation, to induce sterility [40]. In the SIT, sterility can be defined as any complete or partial structural or functional failure to produce gametes or viable zygotes that can be induced by ionizing energy such as gamma irradiation [49,50]. Based on the data presented here, density, temperature, and duration probably only affect somatic cells, thus affecting the physical and fitness of gamma-sterilized male *Ae*. *aegypti,* but not affecting gonad cells to induce sterility. Our findings are similar to those of a previous study conducted by Sasmita et al. [51], who reported that packing and transportation treatment did not affect the induced sterility of gamma-sterilized male *Ae. aegypti*.

The longevity of sterile male mosquitoes post-packaging and simulated transportation treatments is correlated to survivability and mating performance and is important for the success of SIT programs [52]. In the present study, the longevity of gamma-sterilized male *Ae. aegypti* was affected by packaging and transportation treatments and showed a significant reduction with increasing density, temperature, and duration; however, there was no difference between the densities of 40 and 80 males/2 mL. Based on our results, temperatures of 7 and 14 °C and a density of 80 males/2 mL may comprise the optimal conditions for transporting gamma-sterilized male *Ae*. *aegypti,* as these conditions maintained a mean longevity ranging from 4.24 ± 0.29 to 5.89 ± 0.42 days (Appendix A). Lower temperatures immobilized male *Ae. aegypti* and minimized physical damage, resulting in the maintenance of longevity post-transportation. Our findings are consistent with those of previous studies showing that lower temperatures during transportation could maintain the survival and longevity of mosquitoes, including *Ae. aegypti* at temperatures ranging from 7 to 14 °C [41,43], *Ae*. *albopictus* at temperatures ranging from 8 to 14 °C [45], and *An*. *arabiensis* at temperatures ranging from 4 to 10 °C [42]. Regarding a nonchilled temperature setup, a previous study by Sasmita et al. [51] found that an eight hours of land transportation (temperatures ranging from 22 to 26 °C) could maintain the longevity of gamma-sterilized male *Ae. aegypti* from 5.4 to 9.1 days on average. However, longer durations must be investigated for practical purposes.

We were able to measure the fitness and quality of the males using induced sterility and other factors reflecting physical ability and survival, i.e., flight ability, longevity, and mortality, despite the mating competitiveness index not being present in our data set due to the number of dependent variables and their combinations. Our experimental data revealed that lower temperature setups (7 and 14 °C) during packaging and simulated transportation could maintain the quality of gamma-sterilized male *Ae*. *aegypti,* as assessed through mortality, flight ability, induced sterility, and longevity. We advise that a temperature of 7 °C be used to immobilize and maintain sterile male *Ae. aegypti* for short-term transportation periods (up to 24 h). As a rule, insects are poikilothermic. Hence, their body temperature is influenced by environmental temperature. A chilling temperature treatment during packaging and transportation causes a quiescence condition and a reduced metabolism rate, consequently reducing the growth and developmental rate of gamma-sterilized male *Ae. aegypti* [53]. In addition, this chilling temperature was shown to cause immobilization and to prevent lost energy reserve due to movement inside the packaging device. Moreover, it was shown to prevent friction, which potentially causes physical damage [51]. The knock-out time related to the temperature changes can be immediate. A study on *An. arabiensis* showed that an immediate change of temperature from room temperature to between 2 and 11 °C only took from 12 ± 3.6 to 25 ± 2.5 s for complete immobilization [42].

Compaction treatment could be a breakthrough, considering the space efficiency issue in developing packaging and transportation methods for mosquitoes in SIT programs. From a compaction perspective, the key point is finding an optimal density for packaging and transportation that can maintain the quality of sterile male mosquitoes. In the present study, gamma-sterilized male *Ae. aegypti* were compacted into Falcon tubes with densities of 40, 80, and 120 males/2 mL. The densities in our recent study were not realistic for operational SIT programs. Nevertheless, the results were adequate to prove that the increasing density during transportation was detrimental to the quality of gamma-sterilized male *Ae. aegypti*, except for the induced sterility parameter. Increasing density resulted in higher mortality and reduced flight ability and longevity. However, no differences were found between the densities of 40 and 80 males/2 mL. In combination with temperature and duration factors, we recommend that 80 males/2 mL be selected as the optimal density for compaction treatment during the short-term transportation of sterile male *Ae. aegypti*. Higher densities cause sterile male mosquitoes to the crowd, potentially causing physical damage and consequently reducing quality parameters [41] and mating competitiveness [54]. Meanwhile, lower densities are postulated to create space inside the packaging device, potentially causing physical interactions and friction due to vibration during transportation and consequently inducing physical damage and reducing the quality of gamma-sterilized male *Ae. aegypti*. Our findings are similar to those reported by Chung et al. [41], in that a density of 40 males/cm^3^ maintained the survival of more than 80% packed male *Ae. aegypti* using Falcon tubes at a temperature of 7 °C.

In this study, compaction, temperature, and duration factors were combined in the assessment of the packaging and transportation treatment of gamma-sterilized male *Ae. aegypti*. Based on the data presented here, we found that compaction with a density of 80 males/2 mL, a temperature of 7 °C, and short-term transportation (no longer than 24 h), comprising an appropriate setup for packaging, could maintain the quality of gamma-sterilized male *Ae. aegypti*. Immobilization at chilling temperatures and compaction treatments are beneficial in maintaining the quality and space efficiency of sterile male *Ae. aegypti* in packaging and transportation methods in operational SIT programs.

## 5. Conclusions

The results of experiments of the present study revealed that compaction density, temperature, and duration factors in packaging and simulated transportation treatments significantly affected the quality of gamma-sterilized male *Ae. aegypti* regarding mortality, flight ability, and longevity. We found that a temperature of 7 °C and a compaction density of 80 males/2 mL could maintain the longevity and quality of gamma-sterilized male *Ae*. *aegypti,* with a mortality rate of less than 20%, a flight ability of at least 70%, and an induced sterility of around 98%. Consequently, based on our results, we recommend a temperature of 7 °C, a compaction density of 80 males/2 mL, and short-term transportation (no more than 24 h) as an appropriate treatment for the packaging of sterile male *Ae. aegypti* in SIT programs. However, this study has not been able to answer whether the vibration factor during transportation affects the quality of mosquitoes. Therefore, further study on the effect of vibration during transportation is needed.

## Figures and Tables

**Figure 1 insects-13-00847-f001:**
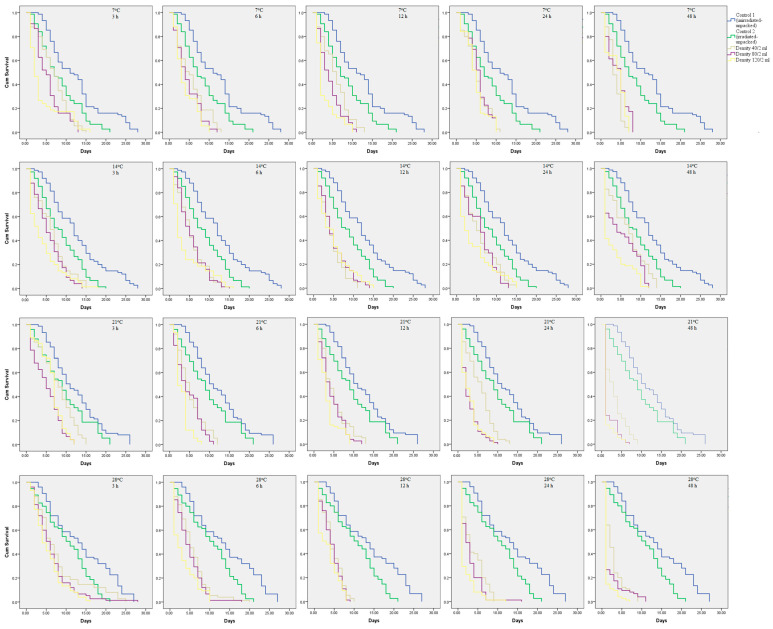
Kaplan–Meier survival curve of gamma-sterilized male *Ae. aegypti* post-treatment. X- and Y-axes represent days and cumulative survival, respectively. Blue, green, grey, purple, and yellow lines represent control 1 (unirradiated–unpacked), control 2 (irradiated–unpacked), and densities of 40, 80, and 120 males/2 mL, respectively.

**Table 1 insects-13-00847-t001:** The GLM analysis results regarding the effects of density, temperature, and duration factors on mortality, flight ability, and induced sterility of gamma-sterilized male *Ae. aegypti*.

Parameter	Factor	df	Mean Square	F	*p*-Value
Mortality	Density	3	447.633	12.123	<0.0001
	Temperature	3	2024.217	54.822	<0.0001
	Duration	4	11,239.344	304.398	<0.0001
	Duration–Density	8	66.429	1.799	0.082
	Duration–Temperature	12	2014.884	54.57	<0.0001
	Temperature–Density	6	362.58	9.82	<0.0001
	Duration–Temperature–Density	24	108.806	2.947	0.001
Flight ability	Density	3	296.245	4.578	0.004
	Temperature	3	7582.074	117.178	<0.0001
	Duration	4	19,054.591	294.482	<0.0001
	Duration–Density	8	413.796	6.395	<0.0001
	Duration–Temperature	12	664.056	10.263	<0.0001
	Temperature–Density	6	392.48	6.066	<0.0001
	Duration–Temperature–Density	24	144.891	2.239	0.001
IS #	Density	2	0.974	0.621	0.539
	Temperature	3	2.836	1.808	0.149
	Duration	4	1.125	0.717	0.582
	Duration–Density	8	0.703	0.448	0.89
	Duration–Temperature	10	1.973	1.258	0.262
	Temperature–Density	6	0.944	0.602	0.728
	Duration–Temperature–Density	20	0.744	0.474	0.972

Notes: # Abbreviation: IS, induced sterility. *p* < 0.05 indicates significance.

**Table 2 insects-13-00847-t002:** Mean mortality of gamma-sterilized male *Ae. aegypti* post-treatment by density, temperature, and duration factors.

Treatments (Density and Duration)		Mean Mortality ± SE (%)			
		7 °C	14 °C	21 °C	28 °C
* Control 1 (unirradiated–unpacked)		0.00 ± 0.00 _aA_	0.42 ± 0.42 _aA_	0.00 ± 0.00 _aA_	0.42 ± 0.42 _aA_
* Control 2 (irradiated–unpacked)		0.83 ± 0.83 _aA_	1.67 ± 0.42 _abA_	1.25 ± 0.72 _aA_	1.67 ± 0.83 _aA_
Density 40 males/2 mL	3 h	20.0 ± 5.2 _abAα_	16.67 ± 3.63 _abcAα_	9.17 ± 3.63 _aAα_	7.5 ± 1.44 _aAα_
	6 h	20.83 ± 8.33 _abAα_	13.33 ± 3.63 _abcAα_	7.5 ± 2.89 _aAα_	7.5 ± 1.44 _aAα_
	12 h	25.83 ± 5.83 _bBα_	20.83 ± 4.41 _bcABα_	7.5 ± 1.44 _aAα_	17.5 ± 2.89 _bABα_
	24 h	17.5 ± 3.82 _abAα_	20.83 ± 7.26 _bcABα_	10.0 ± 2.89 _aAα_	40.83 ± 2.2 _cBα_
	48 h	32.5 ± 6.29 _bAα_	22.5 ± 3.82 _cAα_	84.17 ± 5.07 _bBα_	68.33 ± 2.2 _dBα_
* Control 1 (unirradiated–unpacked)		0.00 ± 0.00 _aA_	0.42 ± 0.42 _aA_	0.00 ± 0.00 _aA_	0.42 ± 0.42 _aA_
* Control 2 (irradiated–unpacked)		0.83 ± 0.83 _abA_	1.67 ± 0.42 _aA_	1.25 ± 0.72 _aA_	1.67 ± 0.83 _abA_
Density 80 males/2 mL	3 h	5.0 ± 2.6 _abAα_	17.92 ± 3.97 _bBα_	10.42 ± 1.1 _abABα_	4.58 ± 0.83 _abAα_
	6 h	14.58 ± 1.1 _cdAα_	15.42 ± 4.23 _bAα_	17.5 ± 0.72 _bAα_	13.75 ± 0.72 _bcAβ_
	12 h	9.17 ± 3.41 _bAα_	17.5 ± 1.91 _bAα_	8.75 ± 2.6 _abAα_	17.5 ± 0.72 _cAα_
	24 h	16.67 ± 0.83 _cdAα_	16.67 ± 1.1 _bAα_	36.67 ± 1.82 _cBβ_	35.42 ± 1.1 _dBα_
	48 h	18.33 ± 1.8 _dAα_	38.75 ± 1.25 _cAβ_	87.5 ± 6.29 _dBα_	80.83 ± 6.47 _eBα_
* Control 1 (unirradiated–unpacked)		0.00 ± 0.00 _aA_	0.42 ± 0.42 _aA_	0.00 ± 0.00 _aA_	0.42 ± 0.42 _aA_
* Control 2 (irradiated–unpacked)		0.83 ± 0.83 _aA_	1.67 ± 0.42 _aA_	1.25 ± 0.72 _aA_	1.67 ± 0.83 _aA_
Density 120 males/2 mL	3 h	25.0 ± 10.49 _bAα_	24.44 ± 0.73 _bAα_	8.33 ± 1.27 _abAα_	6.11 ± 1.0 _aAα_
	6 h	13.06 ± 2.42 _abABα_	22.78 ± 2.37 _bBCα_	9.72 ± 2.82 _abAα_	32.78 ± 1.47 _bCɣ_
	12 h	16.94 ± 5.05 _abAα_	21.94 ± 2.27 _bABα_	22.5 ± 1.44 _bABβ_	31.11 ± 0.73 _bBβ_
	24 h	9.44 ± 1.94 _abAα_	27.78 ± 0.73 _bABα_	28.89 ± 9.69 _bABαβ_	50.0 ± 0.96 _cBβ_
	48 h	29.72 ± 4.34 _bAα_	38.89 ± 2.82 _cAβ_	85.56 ± 5.3 _cBα_	86.94 ± 3.38 _dBα_

* Control 1 and control 2 were not packed and exposed to temperature treatment. The same lowercase, uppercase, and symbol indicate no significant difference within the same density and temperature, the same duration and density, and the same duration and temperature, respectively (one-way ANOVA post hoc Tukey, *p* = 0.05).

**Table 3 insects-13-00847-t003:** Mean flight ability of gamma-sterilized male *Ae. aegypti* post-treatment by density, temperature, and duration factors.

Treatments (Density and Duration)		Mean Flight Ability ± SE (%)			
		7 °C	14 °C	21 °C	28 °C
* Control 1 (unirradiated–unpacked)		100.00 ± 0.00 _cA_	99.38 ± 0.36 _dA_	99.69 ± 0.31 _bA_	98.75 ± 0.88 _dA_
* Control 2 (irradiated–unpacked)		99.06 ± 0.31 _cA_	99.06 ± 0.6 _dA_	99.06 ± 0.6 _bA_	98.44 ± 0.31 _dA_
Density 40 males/2 mL	3 h	99.38 ± 0.63 _cBα_	93.13 ± 3.59 _cdABα_	82.5 ± 5.1 _bAα_	77.5 ± 4.79 _cAα_
	6 h	96.25 ± 1.61 _cBβ_	86.25 ± 1.61 _cdBαβ_	82.5 ± 5.1 _bBα_	54.38 ± 0.05 _bAα_
	12 h	85.63 ± 3.13 _bcBα_	82.5 ± 4.89 _bcBα_	77.5 ± 4.89 _bBα_	39.38 ± 0.07 _abAα_
	24 h	73.75 ± 5.05 _abBα_	70.63 ± 3.44 _abBα_	38.75 ± 9.71 _aAα_	38.13 ± 0.03 _abAα_
	48 h	59.38 ± 6.16 _aBβ_	57.5 ± 4.56 _aBβ_	28.13 ± 6.07 _aAα_	31.25 ± 0.04 _aAα_
* Control 1 (unirradiated–unpacked)		100.00 ± 0.00 _dA_	99.38 ± 0.36 _cA_	99.69 ± 0.31 _cA_	98.75 ± 0.88 _eA_
* Control 2 (irradiated–unpacked)		99.06 ± 0.31 _dA_	99.06 ± 0.6 _cA_	99.06 ± 0.6 _cA_	98.44 ± 0.31 _eA_
Density 80 males/2 mL	3 h	100.0 ± 0.0 _dBα_	92.5 ± 0.72 _cBα_	78.13 ± 3.63 _bcAα_	80.31 ± 0.04 _dAα_
	6 h	89.69 ± 1.87 _cBCα_	90.94 ± 3.73 _cCβ_	78.44 ± 3.08 _bcBα_	66.56 ± 0.02 _cdAα_
	12 h	78.13 ± 1.49 _bBα_	65.0 ± 8.37 _bABα_	81.25 ± 5.54 _bcBα_	48.44 ± 0.03 _abAα_
	24 h	79.06 ± 2.25 _bAα_	72.81 ± 1.93 _bAα_	62.19 ± 9.46 _bAα_	57.19 ± 0.05 _bcAβ_
	48 h	23.44 ± 3.83 _aAα_	34.06 ± 3.76 _aAα_	23.13 ± 4.75 _aAα_	34.69 ± 0.06 _aAα_
* Control 1 (unirradiated–unpacked)		100.00 ± 0.00 _dA_	99.38 ± 0.36 _cA_	99.69 ± 0.31 _cA_	98.75 ± 0.88 _dA_
* Control 2 (irradiated–unpacked)		99.06 ± 0.31 _dA_	99.06 ± 0.6 _cA_	99.06 ± 0.6 _cA_	98.44 ± 0.31 _dA_
Density 120 males/2 mL	3 h	99.79 ± 0.21 _dBα_	90.63 ± 1.97 _cBα_	70.83 ± 3.45 _bAα_	70.0 ± 0.02 _cAα_
	6 h	88.75 ± 1.3 _cAα_	69.58 ± 7.05 _bAα_	80.0 ± 4.99 _bcAα_	69.38 ± 0.05 _cAα_
	12 h	78.13 ± 2.49 _bBα_	76.04 ± 1.29 _bBα_	75.0 ± 5.35 _bBα_	44.38 ± 0.03 _bAα_
	24 h	76.04 ± 1.38 _bBα_	65.21 ± 1.57 _bBα_	39.38 ± 6.98 _aAα_	40.83 ± 0.02 _abAα_
	48 h	34.38 ± 1.2 _aAα_	35.63 ± 1.91 _aAα_	32.71 ± 4.2 _aAα_	28.54 ± 0.02 _aAα_

* Control 1 and control 2 were not packed and exposed to temperature treatment. The same lowercase, uppercase, and symbol indicate no significant difference within the same density and temperature, the same duration and density, and the same duration and temperature, respectively (one-way ANOVA post hoc Tukey, *p* = 0.05).

**Table 4 insects-13-00847-t004:** Mean induced sterility of gamma-sterilized male *Ae. aegypti* post-treatment by density, temperature, and duration factors.

Treatments (Density and Duration)		Mean Induced Sterility ± SE (%)			
		7 °C	14 °C	21 °C	28 °C
* Control 1 (unirradiated–unpacked)		-	-	-	-
* Control 2 (irradiated–unpacked)		97.55 ± 2.18 _aA_	98.5 ± 0.64 _aA_	97.4 ± 0.33 _aA_	98.51 ± 0.66 _aA_
Density 40 males/2 mL	3 h	98.41 ± 0.49 _aAα_	98.0 ± 0.75 _aAα_	98.86 ± 0.61 _aAα_	98.88 ± 0.35 _aAα_
	6 h	98.17 ± 0.72 _aAα_	98.3 ± 0.59 _aAα_	99.17 ± 0.47 _aAα_	99.37 ± 0.26 _aAα_
	12 h	98.68 ± 0.45 _aAα_	98.13 ± 0.78 _aAα_	98.27 ± 0.81 _aAα_	98.63 ± 0.55 _aAα_
	24 h	98.79 ± 0.26 _aAα_	98.159 ± 0.92 _aAα_	99.52 ± 0.24 _aAα_	96.7 ± 1.85 _aAα_
	48 h	98.92 ± 0.57 _aAα_	95.34 ± 2.27_aAα_	n/a	n/a
* Control 1 (unirradiated–unpacked)		-	-	-	-
* Control 2 (irradiated–unpacked)		97.55 ± 2.18 _aA_	98.5 ± 0.64 _aA_	97.4 ± 0.33 _aA_	98.51 ± 0.66 _aA_
Density 80 males/2 mL	3 h	98.04 ± 0.31 _aAα_	98.34 ± 0.28 _aAα_	98.77 ± 0.43 _aAα_	98.87 ± 0.42 _aAα_
	6 h	98.16 ± 0.6 _aAα_	98.42 ± 0.73 _aAα_	98.99 ± 0.42 _aAα_	98.2 ± 0.47 _aAα_
	12 h	98.29 ± 0.09 _aAα_	98.56 ± 0.59 _aAα_	98.28 ± 0.67 _aAα_	98.08 ± 0.67 _aAα_
	24 h	98.7 ± 0.6 _aAα_	98.69 ± 0.27 _aAα_	98.61 ± 0.58 _aAα_	98.99 ± 0.53 _aAα_
	48 h	98.44 ± 0.29 _aAα_	98.15 ± 1.52 _aAα_	n/a	n/a
* Control 1 (unirradiated–unpacked)		-	-	-	-
* Control 2 (irradiated–unpacked)		97.55 ± 2.18 _aA_	98.5 ± 0.64 _aA_	97.4 ± 0.33 _aA_	98.51 ± 0.66 _aA_
Density 120 males/2 mL	3 h	98.29 ± 0.4 _aAα_	98.65 ± 0.31 _aAα_	98.81 ± 0.31 _aAα_	99.02 ± 0.5 _aAα_
	6 h	98.46 ± 0.35 _aAα_	98.31 ± 0.3 _aAα_	99.02 ± 0.48 _aAα_	98.81 ± 0.64 _aAα_
	12 h	98.39 ± 0.56 _aAα_	98.36 ± 0.35 _aAα_	98.44 ± 0.74 _aAα_	98.59 ± 0.54 _aAα_
	24 h	98.43 ± 0.56 _aAα_	98.41 ± 0.46 _aAα_	99.23 ± 0.24 _aAα_	98.33 ± 0.62 _aAα_
	48 h	99.08 ± 0.6 _aAα_	97.41 ± 0.57 _aAα_	n/a	n/a

* Control 1 and control 2 were not packed and exposed to temperature treatment. The same lowercase, uppercase, and symbol indicate no significant difference within the same density and temperature, the same duration and density, and the same duration and temperature, respectively (one-way ANOVA post hoc Tukey, *p* = 0.05). No data were applicable for the duration of 48 h and temperatures of 21 and 28 °C in all densities due to the high mortality rate (>80%).

## Data Availability

The data presented in this study are available in the article and Appendix A.

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
