# Peer review of "Assessment of Compaction, Temperature, and Duration Factors for Packaging and Transporting of Sterile Male Aedes aegypti (Diptera: Culicidae) under Laboratory Conditions"

_insects, 2022, doi:10.3390/insects13090847_

Round 1

Reviewer 1 Report

Due to the complexity of the parameters you have considered in the study, it results some time difficult to follow the data in the figures. I therefore suggested to put the data in tables. Please, see also some comments/suggestions in the text.

Author Response

We want to thank Reviewer 1 for the valuable comments which helped us improve the manuscript. We agree to present the data including posthoc tests in Tables to facilitate the data interpretation to be more informative. Figures 1, 2, and 3 representing mortality, flight ability, and induced sterility, respectively, have been changed to Tables.

For the specific response, please see the attachment.

Reviewer 2 Report

Thank you for inviting me to review the ms. It is an interesting work that would benefit from more than one repetition (three replicates) and some English proof-editing to correct some typos. In addition, authors should be careful in interpreting single factor analysis outputs when there is a significant effect of the interaction of several factors. Sharing the raw data would better help advising on the results. Authors will find some comments/suggestions below and on the marked-up pdf that may improve the ms.

Author Response

We would like to thank Reviewer 2 for the valuable comments which helped us improve the manuscript. We agree to present the data including posthoc tests in Tables to facilitate the data interpretation to be more informative. Figures 1, 2, and 3 representing mortality, flight ability, and induced sterility, respectively, have been changed to Tables.

For the specific response, please see attachment.

Reviewer 3 Report

The authors have carried out a massive and complex experiment in order to obtain conclusions of great interest for the development of improvements in the application of the Sterile Insect Technique in mosquitoes, particularly Ae. aegypti. The experimental design is generally sound, although a third control, with vibration, would have been interesting. The problem, in my opinion, comes from the data treatment and its interpretation. Especially, for the mortality and flight ability parameters, they merely carry out an analysis of variance, determining that there are significant differences among the levels of each factor, as well as interactions among these factors. They do not present results that allow a numerical comparison among the different levels of the factors, beyond a graph that is saturated with information and difficult to interpret. In terms of the objectives of the experiment, it is relevant to know if there are differences among the different levels of a given factor, and how big they are. In fact, their main conclusion (the optimal setup) should be supported on this kind of comparisons. Although they mention having performed post-hoc comparisons, they do not present these results. Their choice of the optimal setup seems to be derived from general observation of the data, as no specific numerical values are mentioned to support their conclusion, let alone statistical pairwise comparisons. This gives the reader the impression that the choice of the optimal setup is somewhat subjective. I will not recommend the publication unless this issue is resolved. Specifically, I suggest you provide statistical evidence for the following statements that are repeated in the text:

-The densities of 40 and 80 are not different from each other (and in turn different from 120) concerning the measured quality parameters.

-The temperature of 7ºC (or 7 and 14 sometimes) have different effect compared to the others and gives the best results concerning the measured quality parameters after the control.

-The acceptable duration of transport is 24h or less, concerning the measured quality parameters.

This can be achieved by reporting the Post-hoc comparisons (the authors claimed that performed these analyses) or alternatively confidence intervals for the Estimated Marginal Means (I think SPSS has an option for this) for each of the levels of each one of the three main factors. Any of these two options will allow you to support numerically your conclusions. Of course, you could also comment after the description factor by factor that there are complex interactions between them (as reveals the significance of the interaction terms).

The presentation of the results is difficult to follow. The model structure is complex, with many factors and levels, and the figures are confusing. The ANOVA results are not especially informative. In my opinion, the main interest is in the comparison between the levels of each factor, and there is no easy way to find out whether there is a trend in the current figures. An informative way to show the data might be to present a separate table or graph with the post-hoc or estimated marginal means.

Aditional comments are included in the attached file.

Author Response

We would like to thank Reviewer 3 for the comprehensive and valuable comments which helped us improve the manuscript. We agree that the data presentation using figures is difficult to understand. In the present study, as mentioned in the statistical analysis section, we perform the GLM, ANOVA followed by the posthoc Tukey test to analyze significantly among means for the parameters of mortality, flight ability, and induced sterility. For the longevity parameter, we performed a Kaplan-Meier curve analysis followed by Mantel-Cox log-rank tests. Based on the suggestions of the reviewers, we agree to present the numerical data including statistical evidence in the Tables to facilitate the data interpretation to be more informative. Figures 1, 2, and 3 represent mortality, flight ability, and induced sterility, respectively, which have been changed to Tables.

The statistical evidence presented in the tables will provide the difference or non-difference between the density of 40 and 80 males/2 ml, temperatures and duration concerning the measured parameters in this study.

Please see attachement for the specific response

Round 2

Reviewer 3 Report

The authors have improved the article in the suggested areas, especially with regard to the presentation of the data. The tables supplied provide the missing information on which their conclusions were based. The minor changes suggested have also been made. The only missing detail is that the authors have omitted to explicitly state that the data to which the linear model has been applied meet the necessary requirements (normality of the data and homogeneity of variances). Given that some of the datasets are percentages, and that these do not usually meet these requirements, or at least require transformation, it would be convenient for the authors to address this issue.
